# The Logistic Burr XII Distribution: Properties and Applications to Income Data

**Renata Rojas Guerra** [1] , **Fernando A. Peña-Ramírez** [2,*] and **Gauss M. Cordeiro** [3]

1   Department of Statistics, Centro de Ciências Naturais e Exatas, Universidade Federal de Santa Maria, Santa Maria 97105-900, Brazil; renata.r.guerra@ufsm.br
2   Department of Statistics, Facultad de Ciencias, Universidad Nacional de Colombia, Bogotá 11001, Colombia
3   Department of Statistics, Centro de Ciências Exatas e da Natureza, Universidade Federal de Pernambuco, Recife 50670-901, Brazil; gauss@de.ufpe.br
*   Correspondence: fapenara@unal.edu.co

**Abstract:** We define and study the four-parameter logistic Burr XII distribution. It is obtained by inserting the three-parameter Burr XII distribution as the baseline in the logistic-X family and may be a useful alternative method to model income distribution and could be applied to other areas. We illustrate that the new distribution can have decreasing and upside-down-bathtub hazard functions and that its density function is an infinite linear combination of Burr XII densities. Some mathematical properties of the proposed model are determined, such as the quantile function, ordinary and incomplete moments, and generating function. We also obtain the maximum likelihood estimators of the model parameters and perform a Monte Carlo simulation study. Further, we present a parametric regression model based on the introduced distribution as an alternative to the location-scale regression model. The potentiality of the new distribution is illustrated by means of two applications to income data sets.

**Keywords:** Burr XII distribution; income distribution; logistic-X family; maximum likelihood estimation; moments





## 1. Introduction

New distributions can often result from the introduction of one or more additional shape parameters to an existing lifetime distribution (say, a baseline model). They are the generalized (or generated) G-classes of distributions. According to [1], there are some reasons why the G-classes attract researchers in several areas. One reason might be the computational refinement of symbolic and numerical programming software. It becomes easier to derive some important mathematical and statistical properties. In addition, the structure of the new generators also allows for the exploration of the distribution's tail properties. Another reason is that the extra parameters obtained from the G baseline models have been shown to improve the quality of fit. Ref. [2] also showed that the G-classes might provide better fits than classical distributions for skewed data.

Several generators have been defined as special cases of the transformed-transformer $(T–X)$ method introduced by [3]. This technique allows for the derivation of families of distributions by using any probability density function (pdf) as a generator.

Let $r(t)$ be the pdf of a random variable $T \in [a, b]$ for $-\infty < a < b < \infty$. Let $G(x)$ be the baseline cumulative distribution function (cdf) of a random variable $X$ such that $W[G(x)]$ satisfies the following conditions:

- $W[G(x)] \in [a, b]$;
- $W[G(x)]$ is differentiable and monotonically non-decreasing;
- $W[G(x)] \to a$ when $x \to -\infty$ and $W[G(x)] \to b$ when $x \to +\infty$.

Therefore, the *T–X* family cdf is defined by

$$F(x) = \int_a^{W[G(x)]} r(t)\mathrm{d}t, \tag{1}$$

and its corresponding pdf is given by

$$f(x) = \left\{ \frac{d}{dx} W[G(x)] \right\} r\{W[G(x)]\}.$$

The *T–X* family of distributions can be classified into subfamilies. One subfamily has the same *X* distribution but different *T* distributions, and the other has the same *T* distribution but different *X* distributions. Some functions $W(\cdot)$, such as $W(x) = -\log(1-x), x/(1-x), \log(x/1-x), \log[-\log(x)]$ for $x \in (0,1)$, will also define different subfamilies.

For example, consider $W(x) = G(x)$. If *T* is a beta random variable, we have the beta-generated family pioneered by [4]. The Kumaraswamy generalized family [5] follows when *T* is a Kumaraswamy random variable. The exponentiated logarithmic generated [2] is also an example of *T-X* special model.

We can also refer to [6] for a class of univariate distributions generated by extending the logistic distribution, called the logistic-X class ("LX" for short). The LX family is a special model of the *T-X* family defined by $W(x) = \log\{-\log[1-G(x)]\}$ in Equation (1) by taking a logistic random variable for *T*. The cdf and pdf of *T* are given by (for $t \in R$) $R(t) = (1 + \mathrm{e}^{-\lambda t})^{-1}$ and $r(t) = \lambda\,\mathrm{e}^{-\lambda t}(1 + \mathrm{e}^{-\lambda t})^{-2}$, respectively, where $\lambda > 0$. Thus, the LX family cdf is defined by

$$F(x) = \left[ 1 + \{-\log[1 - G(x)]\}^{-\lambda} \right]^{-1} \tag{2}$$

and its pdf is given by

$$f(x) = \frac{\lambda\,g(x)}{1 - G(x)} \left[ 1 + [-\log(1 - G(x)]^{-\lambda} \right]^{-(\lambda+1)} \left\{ 1 + [-\log(1 - G(x)]^{-\lambda} \right\}^{-2}, \tag{3}$$

where $G(x)$ is any baseline cdf and $g(x) = \mathrm{d}G(x)/\mathrm{d}x$. The LX family has the same parameter as the baseline distribution plus an additional shape parameter $\lambda > 0$. Note that the baseline distribution is not a special case of the LX family. However, it can be interpreted as a compounding model between the logistic and the baseline distributions. According to [6], this family may allow the construction of symmetric, left-skewed, right-skewed, and/or reverse J-shaped distributions; the definition of models with more types of hazard rate function (hrf); and the provision of competitive models to other generated families under the same baseline distribution, among other characterizations.

In this paper, we introduce a new four-parameter distribution called the logistic Burr XII (LBXII) distribution. It is defined by inserting the three-parameter Burr XII (BXII) distribution as the baseline in Equations (2) and (3). The BXII distribution has a cdf and pdf (for $x > 0$) given by

$$G(x; s, d, c) = 1 - \left[ 1 + \left( \frac{x}{s} \right)^c \right]^{-d} \tag{4}$$

and

$$g(x; s, d, c) = c\,d\,s^{-c}\,x^{c-1} \left[ 1 + \left( \frac{x}{s} \right)^c \right]^{-d-1},$$

respectively, where $d > 0$ and $c > 0$ are shape parameters and $s > 0$ is a scale parameter.

The BXII distribution was originally proposed by [7]. The utilization of the BXII distribution is appealing for several reasons. Notably, it possesses the capacity to effectively capture asymmetric behaviors and heavy-tailed distributions in positive outcomes [8].

These characteristics have led to its adoption as a fundamental tool for the development of generalized probability distributions. Table 1 provides a review of some BXII generalizations through different G families or transformation methods. The BXII distribution also finds extensive application in diverse fields, such as remote sensing [9], econometrics [8], and environmetrics [10].

**Table 1.** Some selected works on Burr XII generalizations.

| Distribution | Author(s) |
| --- | --- |
| Exponentiated BXII | Al-Hussaini and Hussein [11] |
| Beta BXII | Paranaíba et al. [12] |
| Kumaraswamy BXII | Paranaíba et al. [13] |
| Marshal–Olkin extended Burr XII | Al-Saiari et al. [14] |
| Beta exponentiated BXII | Mead [15] |
| McDonald BXII | Gomes et al. [16] |
| BXII negative binomial | Ramos et al. [17] |
| Transmuted BXII | Al-Khazaleh [18] |
| Kumaraswamy exponentiated BXII | Mead and Afify [19] |
| Gamma BXII | Guerra et al. [20] |
| Weibull BXII | Afify et al. [21] and Guerra et al. [22] |
| Flexible Weibull BXII | Elbiely and Yousof [23] |
| BXII-BXII | Gad et al. [24] |
| BXII-moment exponential | Bhatti et al. [25] |
| Unit BXII | Korkmaz and Chesneau [26] and Ribeiro et al. [27] |
| Reflected unit BXII | Ribeiro et al. [28] |
| Type II Topp–Leone BXII | Ogunde and Adeniji [29] |
| New modified BXII | Bhatti et al. [30] |
| Odd-log-logistic BXII | Santos and Pescim [31] |

The cdf of the LBXII distribution is given by (for $x > 0$)

$$F(x) = \left\{ 1 + \left[ d \log \left\{ 1 + \left( \frac{x}{s} \right)^c \right\} \right]^{-\lambda} \right\}^{-1}, \tag{5}$$

where $\lambda > 0$, $d > 0$, and $c > 0$ are shape parameters and $s > 0$ is a scale parameter. The corresponding pdf has the form

$$f(x) = \frac{\lambda \, c \, d^{-\lambda} \, s^{-c} x^{c-1}}{1 + (x/s)^c} \left[ \log \left\{ 1 + \left( \frac{x}{s} \right)^c \right\} \right]^{-(\lambda+1)} \left\{ 1 + \left[ d \log \left\{ 1 + \left( \frac{x}{s} \right)^c \right\} \right]^{-\lambda} \right\}^{-2}. \tag{6}$$

Henceforth , if $X$ is a random variable with density function (6), we write $X \sim$ LBXII($c, d, s, \lambda$). Figure 1 displays plots of the LBXII density function for selected parameter values. It can take various forms, and has as special models some well-known distributions. For $d = 1$ and $s = m^{-1}$, we have the logistic-log-logistic (LLL) distribution. For $c = 1$ it becomes the logistic-Lomax (LLo) model. The hrf of $X$ can be expressed as

$$h(x) = \frac{\lambda \, c \, s^{-c} x^{c-1}}{\log[1 + (x/s)^c]} \left[ 1 + \left( \frac{x}{s} \right)^c \right]^{-1} \left\{ 1 + \left[ d \log \left\{ 1 + \left( \frac{x}{s} \right)^c \right\} \right]^{-\lambda} \right\}^{-1}.$$

Figure 2 provides plots of the hrf for some parameter values. It reveals that the LBXII distribution can have decreasing and upside-down-bathtub hazard functions. The proposed distribution is quite flexible regarding the pdf and hrf and may be a useful alternative to the BXII model and its generalizations. Therefore, it can be considered for modeling income distribution and also in actuarial science, bioscience, and lifetime data, among other areas.

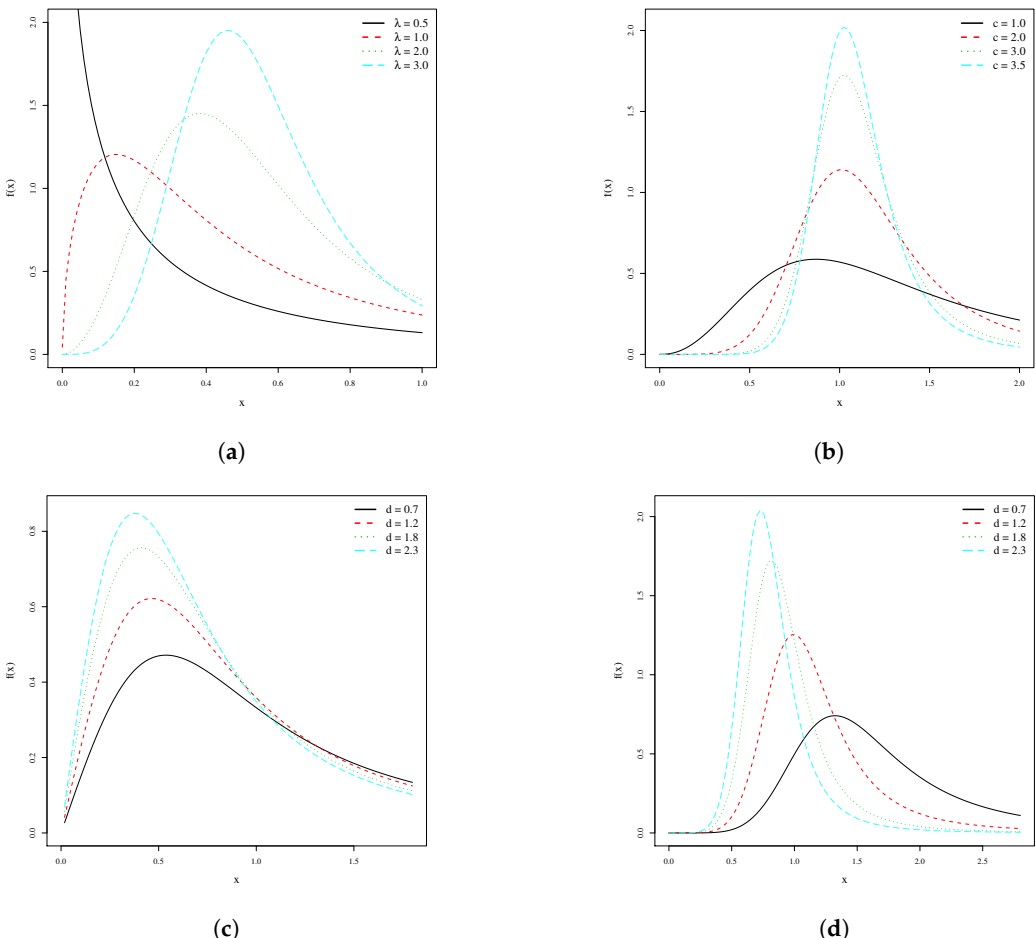

(**a**)

(**b**)

(**c**)

(**d**)

**Figure 1.** Plots of the LBXII density for $s = 1$. (**a**) $c = 1.5$ and $d = 3.0$; (**b**) $\lambda = 3.5$ and $d = 1.2$; (**c**) $\lambda = 0.8$ and $c = 2.5$; (**d**) $\lambda = 3.0$ and $c = 2.5$.

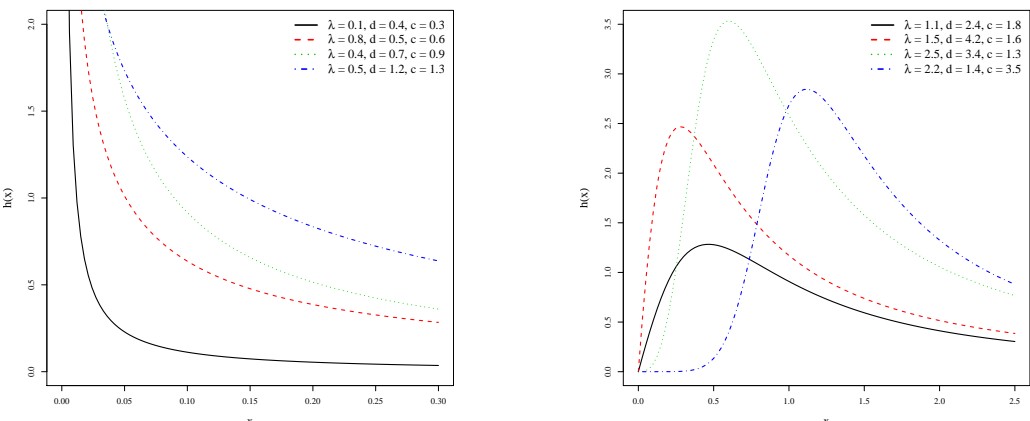

**Figure 2.** Plots of the LBXII hrf for $s = 1$.

The rest of the paper is organized as follows: We derive useful expansions for the cdf and pdf of the new distribution in Section 2. In Section 3, some mathematical properties of the LBXII distribution are investigated. In Section 4, the maximum likelihood method is presented to estimate the model parameters. A simulation study is performed in Section 5. In Section 6, we illustrate the flexibility of the new model using two real data sets. Some concluding remarks are offered in Section 7.

## 2. Useful Expansions

Tahir et al. [6] demonstrated that the LX pdf can be written as an infinite linear combination of exponentiated-G (exp-G) densities; see [32] for the definition of the exp-G distribution. In this section, we derive useful expansions for the LBXII pdf not from exponentiated models but based on our baseline model. Inserting (4) into Equation (2), the LBXII cdf can be rewritten as

$$F(x) = \frac{1}{1 + \left[ -\log\left(1 - \left\{1 - \left[1 + (x/s)^c\right]^{-d}\right\}\right)\right]^{-\lambda}}. \tag{7}$$

Using the Mathematica software, version 12.0, we obtain a power series for $w = 1 + [-\log(1 - y)]^a$ as

$$\begin{aligned} w = 1 + &\left[ 1 + \frac{a}{2} y + \frac{1}{24}(3a^2 + 5a) y^2 + \frac{1}{48}(a^3 + 5a^2 + 6a) y^3 \right. \\ &\left. + \frac{1}{5760}(15a^4 + 150a^3 + 485a^2 + 502a) y^4 \right] y^a + O(y^{a+5}). \end{aligned}$$

Applying this power series for $y = 1 - \left[1 + (x/s)^c\right]^{-d}$ in (7) and after some algebraic manipulation, we obtain

$$F(x) = \frac{\left\{1 - \left[1 + (x/s)^c\right]^{-d}\right\}^\lambda}{\left\{1 - \left[1 + (x/s)^c\right]^{-d}\right\}^\lambda + \sum_{k=0}^\infty p_k \left\{1 - \left[1 + (x/s)^c\right]^{-d}\right\}^k}, \tag{8}$$

where the $p_k$'s are $p_0 = 1, p_1 = \lambda/2, p_2 = \lambda(3\lambda + 5)/24, p_3 = \lambda(\lambda^2 + 5\lambda + 6)/48, p_4 = \lambda(15\lambda^3 + 150\lambda^2 + 485\lambda + 502)/5760$, etc. For any $\lambda > 0$ real non-integer, the following expansion holds since the left-hand-side expression is a cdf

$$\{1 - [1 + (x/s)^c]\}^\lambda = \sum_{k=0}^\infty q_k \{1 - [1 + (x/s)^c]\}^k,$$

where

$$q_k = \sum_{j=k}^\infty (-1)^{k+j} \binom{\lambda}{j}\binom{j}{k}.$$

Thus, Equation (8) can be rewritten as

$$F(x) = \frac{\sum_{k=0}^\infty q_k \{1 - [1 + (x/s)^c]\}^k}{\sum_{k=0}^\infty v_k \{1 - [1 + (x/s)^c]\}^k}, \tag{9}$$

where $v_k = q_k + p_k$. The coefficients of the quotient of the two power series in (9) can be determined from the recurrence equation (for $k \geq 0$)

$$\omega_k = \frac{1}{v_0}\left(q_k - \frac{1}{v_0}\sum_{l=0}^k v_r\,\omega_{k-l}\right)$$

and then, Equation (9) reduces to

$$F(x) = \sum_{k=0}^\infty \omega_k\,H_k(x), \tag{10}$$

where $H_k(x) = \left\{1 - \left[1 + \left(\frac{x}{s}\right)^c\right]^{-d}\right\}^k$. By differentiating (10), we obtain

$$f(x) = \sum_{k=0}^{\infty} \omega_{k+1} h_{k+1}(x)$$
$$= \omega_1 g(x; s, d, c)$$
$$+ \sum_{k=1}^{\infty} \omega_{k+1} (k+1) c d s^{-c} x^{c-1} \left[1 + \left(\frac{x}{s}\right)^c\right]^{-d-1} \left\{1 - \left[1 + \left(\frac{x}{s}\right)^c\right]^{-d}\right\}^k, \quad (11)$$

where $h_{k+1}(x)$ is the exp-BXII pdf with power parameter $k+1$. Using the binomial theorem (for $k \geq 1$), we can write

$$\left\{1 - \left[1 + (x/s)^c\right]^{-d}\right\}^k = \sum_{r=0}^{k} (-1)^r \binom{k}{r} \left[1 + (x/s)^c\right]^{-rd}. \quad (12)$$

Inserting (12) into Equation (11) and after some algebra, we obtain

$$f(x) = \sum_{k=0}^{\infty} \sum_{r=0}^{k} \frac{(-1)^r (k+1) \omega_{k+1}}{r+1} \binom{k}{r} g(x; s, (r+1)d, c),$$

where $g(x; s, (r+1)d, c)$ is the BXII density function with scale parameter $s$ and shape parameters $c$ and $(r+1)d$. Since the sums in the above expressions vary in equal sets of indices, we can exchange $\sum_{k=0}^{\infty} \sum_{r=0}^{k}$ for $\sum_{r=0}^{\infty} \sum_{k=r}^{\infty}$. Therefore, the LBXII pdf can be reduced to

$$f(x) = \sum_{r=0}^{\infty} \rho_r g(x; s, (r+1)d, c), \quad (13)$$

where

$$\rho_r = \sum_{k=r}^{\infty} \frac{(-1)^r (k+1) \omega_{k+1}}{r+1} \binom{k}{r}.$$

Equation (13) is the main result of this section. So, the LBXII pdf is an infinite linear combination of BXII densities. Thus, some mathematical properties of $X$ can be derived from those BXII properties.

## 3. Mathematical Properties

In this section, we obtain some structural properties of the LBXII distribution by establishing algebraic expansions. This might be better than computing those directly by numerical integration of the density function of $X$. We obtain the quantile function, ordinary and incomplete moments, mean deviations, Bonferroni and Lorenz curves, and moment-generating function (mgf).

### 3.1. Quantile Function

The quantile function (qf) of $X$ is determined by inverting (5). We have

$$Q(u) = s \left[\exp\left\{\frac{1}{d}\left(\frac{1}{u} - 1\right)^{-\frac{1}{\lambda}}\right\} - 1\right]^{\frac{1}{c}}. \quad (14)$$

If $U$ has the uniform distribution in $(0, 1)$, the random variable $X = Q(U)$ has the LBXII distribution. Thus, simulating the random variable $X$ is straightforward by using the inverse transform method. We can also have any quantiles of interest by setting appropriate values of $u$. For example, $u = 1/2$ in (14) gives the median $M$ of $X$.

Further, we have alternative expressions for the skewness and kurtosis coefficients based on quantile measures that can be obtained from (14). The Bowley's skewness [33] is given by

$$B = \frac{Q(3/4) - 2Q(1/2) + Q(1/4)}{Q(3/4) - Q(1/4)}.$$

The Moors' kurtosis [34] is defined by

$$M = \frac{Q(7/8) - Q(5/8) + Q(3/8) - Q(1/8)}{Q(6/8) - Q(2/8)}.$$

Some plots of $B$ and $M$ are displayed in Figure 3. They reveal the variation in these measures for different shape parameters.

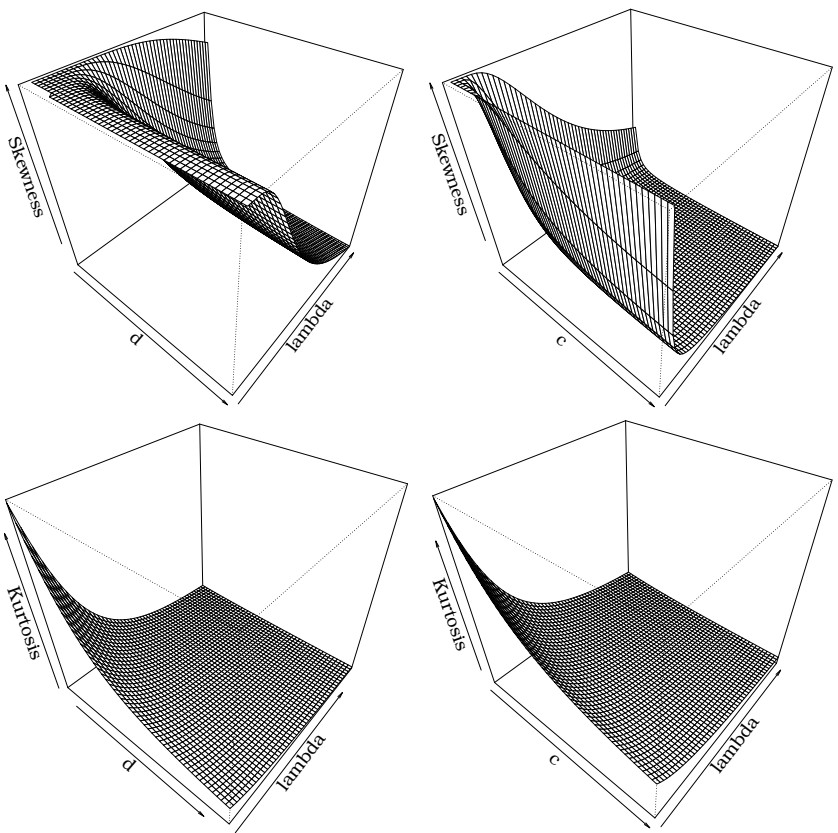

**Figure 3.** Skewness and kurtosis of $X$ for some parameter values.

### 3.2. Ordinary Moments

A result from [35] gives the $h$th ordinary moment of the BXII distribution (for $h < c\,d$) as

$$\mu'_h = s^h\,d\,B(d - h\,c^{-1}, 1 + h\,c^{-1}),$$

where $B(a,b) = \int_0^1 t^{a-1}\,(1-t)^{b-1}dt$ is the beta function. Thus, the $h$th ordinary moment of $X$ can be expressed directly from (13) as (for $h < c\,d$)

$$\mu'_h = s^h\,d\,\sum_{r=0}^{\infty}(r+1)\rho_r\,B((r+1)d - h\,c^{-1}, 1 + h\,c^{-1}). \tag{15}$$

By setting $h = 1$, we obtain the mean of $X$. The moments are most commonly taken about the mean. These so-called central moments ($\mu_s$) follow recursively from (15) as

$$\mu_s = \sum_{i=0}^{s} \binom{s}{i} (-1)^i \mu_1'^s \mu_{s-i}'.$$

The central cumulants ($\kappa_s$) of $X$ can also be determined recursively as

$$\kappa_s = \mu_s' - \sum_{i=1}^{s-1} \binom{s-1}{i-1} \kappa_i \mu_{s-i}',$$

where $\kappa_1 = \mu_1'$. Thus, $\kappa_2 = \mu_2' - \mu_1'^2$, $\kappa_3 = \mu_3' - 3\mu_2'\mu_1' + 2\mu_1'^3$, $\kappa_4 = \mu_4' - 4\mu_3'\mu_1' - 3\mu_2'^2 + 12\mu_2'\mu_1'^2 - 6\mu_1'^4$, etc.

### 3.3. Incomplete Moments

Let $T_h(y) = \int_0^y x^h f(x) dx$ be the $h$th incomplete moment of $X$. It can be derived using the linear representation (13) as

$$T_h(y) = c\,d \sum_{r=0}^{\infty} (r+1)\rho_r \int_0^y x^{h-1} \left(\frac{x}{s}\right)^c \left[1 + \left(\frac{x}{s}\right)^c\right]^{-(r+1)d-1} dx. \qquad (16)$$

Setting $t = \left[1 + \left(\frac{x}{s}\right)^c\right]^{-1}$ in the last equation, we have

$$T_h(y) = d\,s^h \sum_{r=0}^{\infty} (r+1)\rho_r \int_{s^c/(s^c+y^c)}^{1} t^{(r+1)d - \frac{h}{c} - 1} (1-t)^{\frac{h}{c}} dt.$$

Hence, the $h$th incomplete moment of $X$ reduces to

$$T_h(y) = d\,s^h \sum_{r=0}^{\infty} (r+1)\rho_r \, B_{s^c/s^c+y^c}\left((r+1)d - h\,c^{-1}, 1 + h\,c^{-1}\right),$$

where $B_z(a,b) = \int_z^1 t^{a-1}(1-t)^{b-1} dt$ is the upper incomplete beta function. By setting $h = 1$, we obtain the first incomplete moment of $X$. Alternatively, taking $u = (x/s)^c$ in Equation (16), we can write

$$T_h(y) \;=\; d\,s^{h-1} \sum_{r=0}^{\infty} (r+1)\rho_r \int_0^{\left(\frac{y}{s}\right)^c} u^{h/c}(1+u)^{-(r+1)d-1} du.$$

The following integral (for $y > -1$ and $a > -1$) is calculated using `Mathematica`

$$J(y, a, b) = \int_0^y z^a (z+1)^{-b} dz$$

$$= \frac{y^{a+1}\,{}_2F_1(a+1, b; a+2; -y)}{a+1},$$

where ${}_2F_1$ is the hypergeometric function defined by

$${}_2F_1(a, b; c; x) = \sum_{k=0}^{\infty} \frac{(a)_k (b)_k}{(c)_k} \frac{x^k}{k!},$$

where $|x| < 1$, $c = 0, -1, -2, \ldots$ and $(z)_n$ is the Pochhammer polynomial. Thus, the $h$th incomplete moment of $X$ can also be written as (for $h < c\,d$)

$$T_h(y) = d\,s^{h-1} \sum_{r=0}^{\infty} (r+1)\rho_r\, J\left(y, \frac{h}{c}, (r+1)d + 1\right).$$

One important application of the first incomplete moment refers to the mean deviations on the mean and the median of $X$. They are given by $\delta_1 = 2\mu_1' F(\mu_1') - 2\,T_1(\mu_1')$ and $\delta_2 = \mu_1' - 2\,T_1(M)$, respectively. The quantity $F(\mu_1')$ is easily obtained from (5), $T_1(\mu_1')$ is the first incomplete moment of $X$ at the mean $\mu_1'$ and $T_1(M)$ at the median $M$. Other useful applications are the Bonferroni and Lorenz curves. For a given probability $\pi$, they are defined by $B(\pi) = T_1(q)/(\pi\mu_1')$ and $L(\pi) = T_1(q)/\mu_1'$, respectively. The quantity $q = Q(\pi)$ is obtained from (14). These curves are useful in economics for studying income and poverty but can be applied in several other fields.

### 3.4. Generating Function

Let $M_d(t)$ be the mgf of the BXII$(c, d, s)$ distribution. Here, we provide a formula for the mgf $M(t) = \int_{-\infty}^{\infty} e^{t\,x} f(x)dx$ of $X$. Clearly, it can be obtained from (13) as

$$M(t) = \sum_{r=0}^{\infty} (r+1)\rho_r\, M_{(r+1)d}(t), \tag{17}$$

where $M_{(r+1)d}(t)$ is the mgf of the BXII$(c, (r+1)d, s)$ distribution.

Guerra et al. [36] presented the following expansion for the BXII mgf (for $t < 0$)

$$M_d(t) = c\,d \sum_{j=0}^{\infty} \binom{-d-1}{j} \Big[ (-st)^{-(j+1)c}\, \gamma((j+1)c, -st)$$
$$+ (-st)^{(d+j)c}\, \Gamma(-(d+j)c, -st) \Big], \tag{18}$$

where $\gamma(a, z) = \int_0^z t^{a-1}\, e^{-t} dt$ and $\Gamma(a, z) = \int_z^{\infty} t^{a-1}\, e^{-t} dt$ are the lower and upper incomplete gamma functions, respectively. Therefore, for $t < 0$, we combine Equations (17) and (18) to express the mgf of $X$ as

$$\begin{aligned} M(t) &= c\,d \sum_{i=0}^{\infty} (r+1)\rho_r \sum_{j=0}^{\infty} \binom{-(r+1)d-1}{j} \Big[ (-st)^{-(j+1)c}\, \gamma((j+1)c, -st) \\ &+ (-st)^{c[(b+r)d+j]}\, \Gamma(-c[(r+1)d+j], -st) \Big]. \end{aligned}$$

## 4. Parameter Estimation

Various parameter estimation methods are available in the literature, with maximum likelihood and Bayesian techniques standing out. The maximum likelihood method is known for its desirable properties, including the ability to construct confidence intervals. On the other hand, Bayesian methods allow for the integration of prior information, making them valuable for situations with limited data or complex models. This section focuses on estimating the LBXII parameter vector using both maximum likelihood and Bayesian approaches.

### 4.1. Maximum Likelihood Estimation

In this section, we determine the maximum likelihood estimators (MLEs) of the model parameters for the proposed distribution. Let $\boldsymbol{\theta} = (\lambda, s, d, c)^T$ be the vector of the model

parameters of the LBXII($\lambda, s, d, c$) distribution and let $x_1, \ldots, x_n$ be a random sample of size $n$ from this distribution. The log-likelihood function for $\boldsymbol{\theta}$ is given by

$$\ell(\boldsymbol{\theta}) = n \log(\lambda \, c \, s^{-1}) - n \, \lambda \log d - \sum_{i=1}^{n} \log u_i + (c-1)c^{-1} \sum_{i=1}^{n} \log(u_i - 1) \qquad (19)$$

$$- (\lambda + 1) \sum_{i=1}^{n} \log \log u_i - 2 \sum_{i=1}^{n} \log \left[ 1 + (d \, \log u_i)^{-\lambda} \right],$$

where $u_i = 1 + \left( \frac{x_i}{s} \right)^c$. The MLE of $\boldsymbol{\theta}$ can be evaluated numerically by maximizing (19) using the R (`optim` function), SAS (`PROC NLMIXED`), or Ox (sub-routine `MaxBFGS`) programs.

The components of $\boldsymbol{U}(\boldsymbol{\theta})$ are given by

$$U_\lambda(\boldsymbol{\theta}) = n \, \lambda^{-1} - n \, \log d - \sum_{i=1}^{n} \log \log u_i + 2 \sum_{i=1}^{n} \frac{\log[d \, \log u_i]}{1 + (d \, \log u_i)^\lambda},$$

$$U_c(\boldsymbol{\theta}) = n \, c^{-1} + \left[ \frac{c^3 + 2c - 1}{c^2} - 1 \right] \sum_{i=1}^{n} \log(u_i - 1) + c^{-1} \sum_{i=1}^{n} (u_i - 1) \log(u_i - 1) u_i^{-1}$$

$$- (\lambda + 1) \, c^{-1} \sum_{i=1}^{n} \frac{(u_i - 1) \log(u_i - 1)}{u_i \log u_i} + 2 \, \lambda \, c^{-1} \sum_{i=1}^{n} \frac{(u_i - 1) \log(u_i - 1)}{u_i + u_i (d \, \log u_i)^\lambda},$$

$$U_d(\boldsymbol{\theta}) = 2 \, \lambda \, d^{-1} \sum_{i=1}^{n} \frac{1}{1 + (d \log u_i)^\lambda} - \lambda \, n \, d^{-1},$$

and

$$U_s(\boldsymbol{\theta}) = c \, s^{-1} \sum_{i=1}^{n} (u_i - 1) u_i^{-1} \left[ 1 + (\lambda + 1) \sum_{i=1}^{n} \frac{1}{\log u_i} - 2 \, d \, \lambda \sum_{i=1}^{n} \frac{1}{1 + (d \, \log u_i)^\lambda} \right]$$

$$- (n + c - 1) s^{-1}.$$

Setting the score vector $\boldsymbol{U}(\boldsymbol{\theta})$ equal to zero and solving the equations simultaneously yields the MLEs of the four parameters. These equations cannot be solved analytically but there are routines for numerical maximization that may be used. In this paper, we adopt the `AdequacyModel` package in the R statistical computing environment [37]. For interval estimation and testing of hypotheses, we require the asymptotic normality of the MLEs. Under standard regularity conditions, the distribution of $\sqrt{n}(\hat{\lambda} - \lambda, \hat{s} - s, \hat{d} - d, \hat{c} - c)$ can be approximated by a multivariate normal $N_4(0, \boldsymbol{J}(\hat{\boldsymbol{\theta}})^{-1})$ distribution. Here, $\boldsymbol{J}(\boldsymbol{\theta})$ is the observed information matrix given by

$$\boldsymbol{J}(\boldsymbol{\theta}) = -\frac{\partial^2 \, \ell(\boldsymbol{\theta})}{\partial \boldsymbol{\theta} \, \partial \boldsymbol{\theta}^T} = \begin{pmatrix} J_{\lambda\lambda} & J_{\lambda c} & J_{\lambda d} & J_{\lambda s} \\ \cdot & J_{cc} & J_{cd} & J_{cs} \\ \cdot & \cdot & J_{dd} & J_{ds} \\ \cdot & \cdot & \cdot & J_{ss} \end{pmatrix},$$

whose elements can be obtained from the authors upon request.

### 4.2. Bayesian Estimation

To address the complexity of the joint likelihood function, we derive Bayesian estimators for the LBXII parameters using a Markov chain Monte Carlo (MCMC) method.

When working with an observed random sample $x = (x_1, \ldots, x_n)^\top$ drawn from the LBXII$(\lambda, s, d, c)$ distribution, the likelihood function is

$$L(\boldsymbol{\theta}|\boldsymbol{x}) \propto \left( \frac{\lambda\, c\, s^{-c}}{d^\lambda} \right)^n \prod_{i=1}^n \frac{x_i^{c-1}}{1 + (x_i/s)^c} \prod_{i=1}^n \left[ \log\left\{ 1 + \left( \frac{x_i}{s} \right)^c \right\} \right]^{-(\lambda+1)}$$

$$\times \prod_{i=1}^n \left\{ 1 + \left[ d\log\left\{ 1 + \left( \frac{x_i}{s} \right)^c \right\} \right]^{-\lambda} \right\}^{-2}.$$

Therefore, we can perform Bayesian estimation by assuming that the unknown parameters are independent and each follows a gamma distribution, which is denoted as $\theta_i \sim \text{Gamma}(p_i, q_i)$, with $\theta_i$ being part of the parameter vector $\boldsymbol{\theta}$ and $i$ ranging from 1 to 4. Here, the hyperparameters $(p_i, q_i)$ are known and positive values. Then, the joint prior distribution of $\lambda$, $s$, $d$, and $c$ is $p(\lambda, s, d, c) \propto \lambda^{p_1-1} s^{p_2-1} d^{p_3-1} c^{p_4-1} e^{-(q_1\lambda + q_2 s + q_3 d + q_4 c)}$. Hence, the joint posterior pdf can be obtained using Bayes' theorem and it is given by

$$\pi(\boldsymbol{\theta}|\boldsymbol{x}) \propto \lambda^{n+p_1-1} s^{-c\,n+p_2-1} d^{-\lambda\,n+p_3-1} c^{n+p_4-1} \exp\{-(q_1\lambda + q_2 s + q_3 d + q_4 c)\}$$

$$\times \prod_{i=1}^n \frac{x_i^{c-1}}{1 + (x_i/s)^c} \prod_{i=1}^n \left[ \log\left\{ 1 + \left( \frac{x_i}{s} \right)^c \right\} \right]^{-(\lambda+1)} \prod_{i=1}^n \left\{ 1 + \left[ d\log\left\{ 1 + \left( \frac{x_i}{s} \right)^c \right\} \right]^{-\lambda} \right\}^{-2}. \tag{20}$$

To derive the marginal posterior distribution for each element in $\boldsymbol{\theta}$, we must integrate Equation (20). To this aim, we employ the MCMC method to draw posterior samples and infer the marginal distributions. To generate these samples, we first derive the conditional posterior distributions for the unknown parameters of the LBXII distribution, which are given by the equations below:

$$\pi(\lambda|s, d, c, \boldsymbol{x}) \propto \lambda^n d^{-\lambda n} \prod_{i=1}^n \left[ \log\left\{ 1 + \left( \frac{x_i}{s} \right)^c \right\} \right]^{-\lambda} \prod_{i=1}^n \left\{ 1 + \left[ d\log\left\{ 1 + \left( \frac{x_i}{s} \right)^c \right\} \right]^{-\lambda} \right\}^{-2} \tag{21}$$

$$\pi(s|\lambda, d, c, \boldsymbol{x}) \propto s^{-c\,n} \prod_{i=1}^n \left[ \log\left\{ 1 + \left( \frac{x_i}{s} \right)^c \right\} \right]^{-(\lambda+1)} \prod_{i=1}^n \left\{ 1 + \left[ d\log\left\{ 1 + \left( \frac{x_i}{s} \right)^c \right\} \right]^{-\lambda} \right\}^{-2}$$

$$\times \prod_{i=1}^n \frac{x_i^{c-1}}{1 + (x_i/s)^c}, \tag{22}$$

$$\pi(d|\lambda, s, c, \boldsymbol{x}) \propto d^{-\lambda n} \prod_{i=1}^n \left\{ 1 + \left[ d\log\left\{ 1 + \left( \frac{x_i}{s} \right)^c \right\} \right]^{-\lambda} \right\}^{-2}, \tag{23}$$

and

$$\pi(c|\lambda, s, d, \boldsymbol{x}) \propto s^{-c\,n} c^n \prod_{i=1}^n \left[ \log\left\{ 1 + \left( \frac{x_i}{s} \right)^c \right\} \right]^{-(\lambda+1)} \prod_{i=1}^n \left\{ 1 + \left[ d\log\left\{ 1 + \left( \frac{x_i}{s} \right)^c \right\} \right]^{-\lambda} \right\}^{-2}$$

$$\times \prod_{i=1}^n \frac{x_i^{c-1}}{1 + (x_i/s)^c}. \tag{24}$$

From Equations (21)–(24) one can note that the full conditional distributions of $\lambda$, $s$, $d$, and $c$ cannot be expressed as any recognizable density function. Consequently, directly generating $\boldsymbol{\theta}$ from $p(\lambda|\cdot)$, $p(s|\cdot)$, $p(d|\cdot)$, and $p(c|\cdot)$ using standard methods is unfeasible. To derive Bayesian estimates for the unknown parameters, we employ the Metropolis–Hastings (M-H) algorithm [38,39], and following the MCMC sampling procedure outlined below:

1.  Initialize the parameter vector $\boldsymbol{\theta} = (\lambda, s, d, c)$ with starting values, denoted as $\boldsymbol{\theta}^{(0)} = (\lambda^{(0)}, s^{(0)}, d^{(0)}, c^{(0)})$, and set the iteration counter $j = 1$.
2.  Propose new values for $\boldsymbol{\theta}$ as $\boldsymbol{\theta}^* = (\lambda^*, s^*, d^*, c^*)^\top$ by sampling from the proposal distribution: $\theta_i^* \sim N(\hat{\theta}_i, \hat{J}_{i,i})$, where $\theta_i^* \in \boldsymbol{\theta}^*$, and $i = 1, \ldots, 4$. Here, $\hat{J}_{i,i}$ represents the $i$-th element of the main diagonal of the observed Fisher information matrix $J(\hat{\boldsymbol{\theta}})$.

3. Calculate

$$h\left(\theta_i^{(j-1)}, \theta_i^*\right) = \min\left\{1, \frac{\pi\left(\theta_i^*|\boldsymbol{\theta}_{-i}^{(j-1)}, \boldsymbol{x}\right)}{\pi\left(\theta_i^{(j-1)}|\boldsymbol{\theta}_{-i}^{(j-1)}, \boldsymbol{x}\right)}\right\}, \quad i = 1, \ldots, 4,$$

which is the acceptance probability, where $\boldsymbol{\theta}_{-i}^{(\cdot)}$ denotes the vector $\boldsymbol{\theta}^{(\cdot)}$ with its $i$-th element removed, and $\pi(\cdot)$ is as given in Equations (21)–(24).

4. Generate $u_i$ ($i = 1, \ldots, 4$) from the standard uniform distribution. If $u_i < h\left(\theta_i^{(j-1)}, \theta_i^*\right)$, set $\theta_i^{(j)} = \theta_i^*$; otherwise, set $\theta_i^{(j)} = \theta_i^{(j-1)}$.

5. Increment the counter from $j$ to $j + 1$.

6. Return to step 2 if $j < M$, where $M$ is a sufficiently large number indicating convergence; otherwise, exclude the initial $N < M$ samples as burn-in and calculate the Bayesian estimates as

$$\tilde{\theta}_i = \frac{1}{M - N} \sum_{j=N+1}^{M} \theta_i^{(j)}.$$

## 5. Simulation Study

In this section, we conduct a Monte Carlo experiment to investigate some asymptotic properties of the MLEs for the parameters of the LBXII distribution. Based on the LBXII qf, we use the inverse transform method to generate five different combinations of parameters $\lambda, c, d$, and $s$ for the LBXII model. Four sample sizes are considered ($n = 50, 100, 250, 500$) and the number of replications is 10,000. We use the R programming language to maximize the log-likelihood (19) and the code for the maximum likelihood estimation is provided in the Appendix A. Table 2 presents the mean estimates of the MLEs and their root mean squared errors (RMSEs). As expected, the MLEs tend to be closer to the true parameters and the RMSEs decrease when the sample size $n$ increases.

**Table 2.** Monte Carlo simulation results for the LBXII mean estimates and RMSEs.

| $\boldsymbol{\theta}$ | n | Mean | | | | RMSE | | | |
|---|---|---|---|---|---|---|---|---|---|
| | | $\hat{\lambda}$ | $\hat{c}$ | $\hat{d}$ | $\hat{s}$ | $\hat{\lambda}$ | $\hat{c}$ | $\hat{d}$ | $\hat{s}$ |
| $(3, 0.2, 2.5, 5)^\top$ | 50 | 3.270 | 0.243 | 3.029 | 5.259 | 1.616 | 0.142 | 2.130 | 3.507 |
| | 100 | 3.278 | 0.224 | 2.791 | 5.172 | 1.443 | 0.104 | 1.428 | 3.040 |
| | 250 | 3.192 | 0.212 | 2.638 | 5.080 | 1.135 | 0.073 | 0.839 | 2.437 |
| | 500 | 3.155 | 0.206 | 2.566 | 5.048 | 0.918 | 0.056 | 0.561 | 1.984 |
| $(6, 4, 5, 0.5)^\top$ | 50 | 6.656 | 4.094 | 5.534 | 0.535 | 2.069 | 1.305 | 2.458 | 0.157 |
| | 100 | 6.423 | 4.080 | 5.280 | 0.518 | 1.670 | 1.088 | 1.984 | 0.109 |
| | 250 | 6.244 | 4.044 | 5.198 | 0.510 | 1.265 | 0.844 | 1.539 | 0.073 |
| | 500 | 6.137 | 4.044 | 5.119 | 0.505 | 1.013 | 0.693 | 1.227 | 0.055 |
| $(9, 1.7, 5, 0.1)^\top$ | 50 | 9.162 | 1.774 | 5.326 | 0.108 | 1.527 | 0.396 | 1.598 | 0.044 |
| | 100 | 9.054 | 1.752 | 5.210 | 0.104 | 1.224 | 0.300 | 1.270 | 0.028 |
| | 250 | 9.027 | 1.725 | 5.093 | 0.102 | 0.901 | 0.203 | 0.911 | 0.018 |
| | 500 | 8.993 | 1.716 | 5.067 | 0.100 | 0.668 | 0.148 | 0.699 | 0.013 |
| $(10.5, 4.2, 6.5, 0.2)^\top$ | 50 | 10.602 | 4.295 | 6.580 | 0.201 | 1.106 | 0.620 | 1.157 | 0.017 |
| | 100 | 10.539 | 4.263 | 6.540 | 0.200 | 0.881 | 0.474 | 0.955 | 0.012 |
| | 250 | 10.517 | 4.231 | 6.528 | 0.200 | 0.694 | 0.343 | 0.747 | 0.010 |
| | 500 | 10.511 | 4.217 | 6.517 | 0.200 | 0.557 | 0.266 | 0.597 | 0.007 |

**Table 2.** *Cont.*

| $\theta$ | n | Mean | | | | RMSE | | | |
|---|---|---|---|---|---|---|---|---|---|
| | | $\hat{\lambda}$ | $\hat{c}$ | $\hat{d}$ | $\hat{s}$ | $\hat{\lambda}$ | $\hat{c}$ | $\hat{d}$ | $\hat{s}$ |
| | 50 | 14.539 | 0.545 | 0.175 | 5.544 | 1.282 | 0.481 | 0.157 | 1.401 |
| $(14.1, 0.5, 0.1, 5.4)^{\top}$ | 100 | 14.438 | 0.540 | 0.164 | 5.487 | 0.995 | 0.445 | 0.138 | 1.194 |
| | 250 | 14.325 | 0.526 | 0.152 | 5.470 | 0.714 | 0.395 | 0.115 | 0.943 |
| | 500 | 14.255 | 0.522 | 0.143 | 5.466 | 0.551 | 0.352 | 0.100 | 0.798 |

## 6. Applications

In this section, we present two examples to illustrate the potentiality of the LBXII distribution for modeling income data. The first data set consists of the annual salaries of professional hockey players for the season 2012–2013. It has 714 observations in American dollars and is available for download at https://www.usatoday.com/sports/nhl/ (accessed on 17 September 2016).

The second example represents the individual payroll income of 5024 Italian households with positive income. These data are obtained from the Survey of Household Income and Wealth (SHIW) of the Bank of Italy for 2014. The observations are measured in euros.

We fit the LBXII model for both data sets and compare them with six other competitive models. The distributions covered in this comparison include five-parameter BXII generalizations and some special models of our proposal. In what follows, we present the mathematical expressions of the density functions under consideration. These expressions are essential to provide a clear and concise reference for readers to understand the potential competitors of the proposed model. Therefore, they are defined below (for $x > 0$):

- The KwBXII density is given by

$$
\begin{aligned}
f(x) \;=\; & a\,b\,c\,d\,s^{-c}x^{c-1}\left[1 + \left(\frac{x}{s}\right)^c\right]^{-d-1}\left\{1 - \left[1 + \left(\frac{x}{s}\right)^c\right]^{-d}\right\}^{a-1} \times \\
& \left[1 - \left\{1 - \left[1 + \left(\frac{x}{s}\right)^c\right]^{-d}\right\}^a\right]^{b-1},
\end{aligned}
$$

where $a > 0$ and $b > 0$ are shape parameters.

- The BBXII density is given by

$$
f(x) = \frac{c\,d\,x^{c-1}}{s^c\,B(a,b)}\left\{1 - \left[1 + (x/s)^c\right]^{-d}\right\}^{a-1}\left[1 + (x/s)^c\right]^{-(d\,b+1)},
$$

where $a > 0$ and $b > 0$ are shape parameters.

- The BXII density is given in (6).
- The exponentiated Weibull (EW) density [32] is given by

$$
g(t) = \alpha\,\beta\,\lambda\,x^{\alpha-1}\exp(-\lambda\,x^{\alpha})[1 - \exp(-\lambda\,x^{\alpha})]^{\beta-1},
$$

where $\alpha > 0$ and $\beta > 0$ are shape parameters and $\lambda > 0$ is a scale parameter.

- The Weibull (W) density, which arises from the EW density when $\beta = 1$.
- The LL density obtained from the BXII density with $s = m^{-1}$ and $d = 1$.

The statistics considered for these models are the following: the Akaike information criterion (AIC), consistent Akaike information criterion (CAIC), Bayesian information criterion (BIC), Bayesian information criteria Hannan–Quinn information criterion (HQIC), and Kolmogorov–Smirnov (KS). The lower the goodness-of-fit statistics, the better the distribution adjustment to the data. We use the R programming language to obtain the MLEs and goodness-of-fit statistics of the LBXII and all its competitor models.

### 6.1. Hockey Players' Salaries

Table 3 provides a descriptive summary of the hockey players' data. We have a higher value for the standard deviation (SD) and an amplitude of 13,475,000. This indicates that the current data have great variability. The skewness is positive, and the kurtosis is large. Further, the mean and median are not so close. These statistics suggest that hockey players' salaries follow a power law distribution, which is very common in income data sets.

**Table 3.** Descriptive statistics for hockey players' data.

| Mean | Median | SD | Skewness | Kurtosis | Min. | Max. |
|------|--------|-----|----------|----------|------|------|
| 2,450,815.39 | $1.675 \times 10^6$ | 2,112,878 | 1.61 | 3.35 | $5.25 \times 10^5$ | $1.4 \times 10^7$ |

The MLEs and their standard errors for all fitted distributions are listed in Table 4. The Bayes estimates, following the procedure described in Section 4.2, are also included. We note that the parameter estimates are significant for all considered models. Table 5 presents the goodness-of-fit statistics and reveals that the LBXII distribution yields a good adjustment for the hockey players' data. It has the lowest values for all statistics, thus indicating it as a competitive alternative to the classical W, EW, and other BXII generalizations and special models.

**Table 4.** The MLEs and Bayesian estimates of the model parameters and their standard errors for hockey players' data.

| | c | d | s | a | b |
|---|---|---|---|---|---|
| BBXII | 0.6639 | 0.1238 | 6.1887 | 12.3249 | 7.0895 |
| | (0.0459) | (0.0087) | (0.9562) | (0.7336) | (0.5711) |
| KwBXII | 5.3659 | 0.0240 | 2.9941 | 8.1383 | 3.5522 |
| | (0.4790) | (0.0021) | (0.4481) | (0.4410) | (0.2546) |
| | *c* | *d* | *s* | *λ* | |
| LBXII | 0.4665 | 0.1676 | 5.0797 | 13.8981 | |
| | (0.0183) | (0.0069) | (1.5760) | (0.4894) | |
| LBXII * | 0.4834 | 0.1723, | 10.33954 | 17.6727 | |
| | (0.0132) | (0.0046) | (1.2755) | (0.3634) | |
| BXII | 7.7501 | 0.0093 | 3.2532 | | |
| | (0.8155) | (0.0010) | (0.4724) | | |
| | *λ* | *α* | *β* | | |
| EW | 1.5782 | 0.0411 | 8.5690 | | |
| | (0.2285) | (0.0059) | (1.1925) | | |
| W | 10.4164 | 0.0683 | | | |
| | (1.3816) | (0.0018) | | | |
| | *c* | *m* | | | |
| LL | 0.1296 | 12.6759 | | | |
| | (0.0040) | (1.7147) | | | |

\* Bayesian estimates.

**Table 5.** Goodness-of-fit statistics for the models fitted to the hockey players' data. The best results are in boldface.

| | AIC | CAIC | BIC | HQIC | KS |
|---|---|---|---|---|---|
| BBXII | 23,764.9605 | 23,765.0452 | 23,787.8149 | 23,773.7870 | 0.3836 |
| KwBXII | 23,954.2392 | 23,954.3239 | 23,977.0936 | 23,963.0656 | 0.4249 |
| LBXII | **22,660.5691** | **22,660.6256** | **22,678.8527** | **22,667.6303** | **0.1957** |
| BXII | 25,640.6040 | 25,640.6378 | 25,654.3166 | 25,645.8999 | 0.5767 |
| EW | 25,032.8320 | 25,032.8658 | 25,046.5446 | 25,038.1279 | 0.6477 |
| W | 26,436.5128 | 26,436.5297 | 26,445.6546 | 26,440.0434 | 0.8768 |
| LL | 26,192.7199 | 26,192.7368 | 26,201.8617 | 26,196.2505 | 0.7987 |

The three estimated densities with lower values for the goodness-of-fit statistics and the histogram of the data are given in Figure 4. They agree with what was discussed in the descriptive summary and the results in Table 5. Thus, the LBXII model is very competitive with the other fitted distributions and provides a better adjustment for the current data.

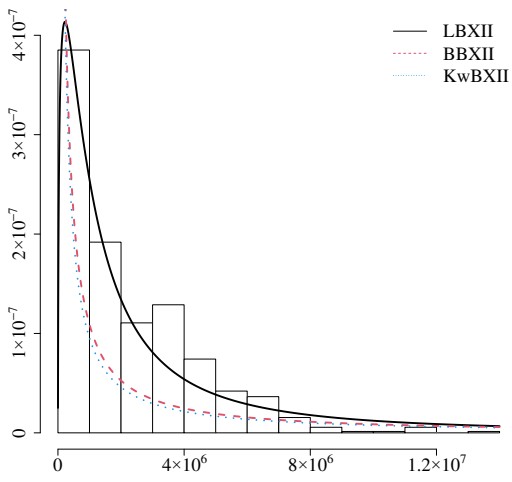

**Figure 4.** Histogram and estimated densities of the LBXII, BBXII, and KwBXII models for hockey players' data.

### 6.2. Individual Payroll Income

Table 6 provides a descriptive summary of the individual payroll income data. For these data, the mean and median are close and the SD is higher. We also note large values for the skewness and kurtosis coefficients. The amplitude is 134,900 for these data. Just like for the first data set, the descriptive statistics indicate that the payroll income may follow a power law distribution with a right-skew tail.

**Table 6.** Descriptive statistics for payroll income data.

| Mean | Median | SD | Skewness | Kurtosis | Min. | Max. |
|---|---|---|---|---|---|---|
| 16,714.67 | 16,200.00 | 9218.184 | 2.62 | 19.29 | 100.00 | 135,000.00 |

Tables 7 and 8 present the MLEs with their standard errors and the goodness-of-fit statistics, respectively, for seven fitted models. The Bayes estimates, following the procedure described in Section 4.2, are also included. These results are obtained for the LBXII distribution and six competitive models. The parameter estimates are significant for all fitted models, and the LBXII distribution exhibits the lowest values for all goodness-of-fit statistics. Similarly to the first empirical example, the LBXII model shows up as a competitive alternative to the other fitted models.

Figure 5 displays a histogram and some plots of the estimated densities for the three most competitive models according to the goodness-of-fit statistics of the payroll income data. These plots are in agreement with the results in Table 8. Similarly to the first data set, the LBXII distribution can be used effectively to provide better fits than other considered income distributions for these data and it is a very competitive alternative to the W and EW models.

**Table 7.** The MLEs and Bayesian estimates of the model parameters and their standard errors for payroll income data.

|          | c        | d        | s        | a        | b        |
|----------|----------|----------|----------|----------|----------|
| BBXII    | 1.8301   | 0.0876   | 8.4148   | 18.3223  | 8.3320   |
|          | (0.0614) | (0.0029) | (0.3872) | (0.4026) | 0.283278 |
| KwBXII   | 5.3594   | 0.0354   | 3.8626   | 8.828    | 5.4490   |
|          | (0.0987) | (0.0007) | (0.2147) | (0.2225) | (0.1341) |
|          | c        | d        | s        | $\lambda$ |         |
| LBXII    | 1.1934   | 0.1057   | 5.3612   | 14.1480  |          |
|          | (0.0310) | (0.0027) | (0.3478) | (0.1928) |          |
| LBXII *  | 1.2242   | 0.1084   | 7.9545   | 14.4076  |          |
|          | (0.0215) | (0.0018) | 0.2933   | 0.1501   |          |
| BXII     | 1.8585   | 0.0812   | 20.4626  |          |          |
|          | (0.1622) | (0.0071) | (0.8067) |          |          |
|          | $\lambda$ | $\alpha$ | $\beta$ |          |          |
| EW       | 2.7063   | 0.0408   | 11.2836  |          |          |
|          | (0.1329) | (0.0020) | (0.4695) |          |          |
| W        | 16.8256  | 0.1169   |          |          |          |
|          | (0.6817) | (0.0013) |          |          |          |
|          | c        | m        |          |          |          |
| LL       | 0.2512   | 32.2073  |          |          |          |
|          | (0.0032) | (1.2346) |          |          |          |

* Bayesian estimates.

**Table 8.** Goodness-of-fit statistics for the models fitted to the payroll income data. The best results are in boldface.

|          | AIC            | CAIC           | BIC            | HQIC           | KS      |
|----------|----------------|----------------|----------------|----------------|---------|
| BBXII    | 112,144.5514   | 112,144.5634   | 112,177.1613   | 112,155.9779   | 0.2825  |
| KwBXII   | 114,423.2534   | 114,423.2654   | 114,455.8633   | 114,434.6799   | 0.3159  |
| LBXII    | **107,006.6913** | **107,006.6992** | **107,032.7792** | **107,015.8325** | **0.2010** |
| BXII     | 125,013.2791   | 125,013.2839   | 125,032.8450   | 125,020.1350   | 0.5063  |
| EW       | 122,416.3162   | 122,416.3210   | 122,435.8822   | 122,423.1721   | 0.5575  |
| W        | 131,889.9303   | 131,889.9327   | 131,902.9743   | 131,894.5009   | 0.8148  |
| LL       | 129,319.9984   | 129,320.0008   | 129,333.0424   | 129,324.5690   | 0.7323  |

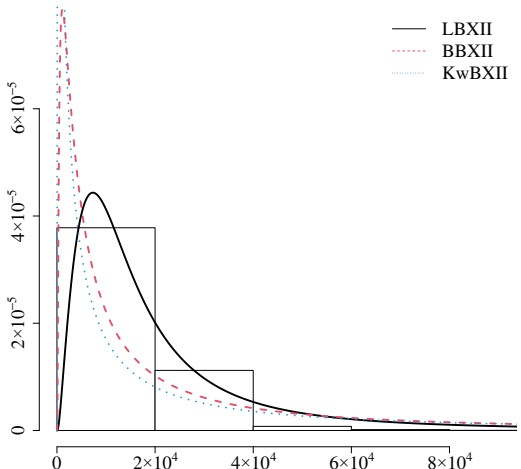

**Figure 5.** Histogram and estimated densities of the LBXII, BBXII, and KwBXII models for payroll income data.

## 7. Concluding Remarks

We introduce the four-parameter *logistic Burr XII* (LBXII) distribution. It can have decreasing and upside-down-bathtub hazard functions and can be considered for modeling income distributions, among other applications. We demonstrate that the LBXII density function is an infinite linear combination of BXII densities. Thus, some mathematical properties of the new distribution are obtained using this result, such as the ordinary and incomplete moments and generating function. We also determine the quantile function for the LBXII distribution, which is useful to obtain any quantiles of interest, simulate LBXII random variables, and provide some alternative expressions for the skewness and kurtosis. We estimate the model parameters using the maximum likelihood method, and a simulation study is provided using a Monte Carlo experiment. In our simulation study, we note that the efficiency of the maximum likelihood estimators improves for larger sample sizes, which is an important aspect to consider when applying the LBXII distribution to real-world data. We present two applications to illustrate the potentiality of the LBXII distribution for modeling income data. Both data sets exhibit characteristics of a power law distribution, which is very common in income data sets. We note that the LBXII distribution has a good adjustment in both cases, thus being a competitive model against the classical Weibull distribution, exponentiated Weibull model, other BXII generalizations, and special models. Finally, the LBXII model may provide an attractive alternative to describe and understand income distribution behavior.

**Author Contributions:** Conceptualization, G.R.R. and F.A.P.-R.; methodology, G.R.R. and F.A.P.-R.; software, G.R.R.; validation, G.R.R.; formal analysis, G.R.R.; investigation, G.R.R. and F.A.P.-R.; resources, G.R.R. and F.A.P.-R.; data curation, G.R.R.; writing—original draft preparation, G.R.R. and F.A.P.-R.; writing—review and editing, F.A.P.-R. and C.G.M.; visualization, G.R.R. and F.A.P.-R.; supervision, C.G.M.; project administration, G.R.R., F.A.P.-R. and C.G.M.; funding acquisition, G.R.R., F.A.P.-R. and C.G.M. All authors have read and agreed to the published version of the manuscript.

**Funding:** This work was supported by the FAPERGS (Fundação de Amparo à pesquisa do Estado do RS) grant numbers and 23/2551-0000851-3; CNPq (Conselho Nacional de Desenvolvimento Científico e Tecnológico) grant number 306274/2022-1.

**Institutional Review Board Statement:** Not applicable.

**Informed Consent Statement:** Not applicable.

**Data Availability Statement:** Data are contained within the article.

**Acknowledgments:** The author, R.R. Guerra, acknowledges the support of Instituto Serrapilheira for its invaluable contribution to her research.

**Conflicts of Interest:** The authors declare no conflict of interest.

## Appendix A. Code for Maximum Likelihood Estimation

This appendix presents the code for the maximum likelihood and Bayesian estimations applied to the proposed distribution. The provided code is designed using the R programming language.

```
# log-likelihood function
fr <- function(par){
c <- par[1]
k <- par[2]
s <- par[3]
l <- par[4]
x=t_i
-(n*(log(l*c*k*s^(-1)))+(c-1)*c^(-1)*
sum(log(1+(x/s)^c-1))-sum(log(1+(x/s)^c))
-(l+1)*(n*log(k)+sum(log(log(1+(x/s)^c))))-2*
sum(log(1+(k*log(1+(x/s)^c))^(-l))))
```

```
}
# setting one scenario
c=3;k=.2;s=2.5;l=5
# setting the sample size
n=500
# generating one sample
set.seed(2023)
u = runif(n=n,min=0,max=1)
t_i=s*(exp(1/k*((1-u)/u)^(-1/l))-1)^(1/c)
##looking for initial values
fit.sa <- function(y,fr) {
minusllike <- function(y) fr(c(y[1],y[2],y[3],y[4]))
lower <- c(0.1,0.1,0.1,0.1) #may need some changes here
upper <- c(10,10,10,10)
out <- GenSA::GenSA(lower = lower, upper = upper,
fn = minusllike, control=list(verbose=F,max.time=2))
return(out[c("value","par","counts")])
}

initial<-fit.sa(y,fr)$par

# maximizing the log-likelihood
res <- optim(initial, fr,  method = "L", lower = 0)
# Maximum likelihood estimators
res$par

#bayesian estimation
library(rstan)
seet<-123
set.seed(seet)
App<-2 #Or 1
est1<-matrix(c( mle1, sd1),2,4,byrow = T)
est2<-matrix(c( mle2, sd2),2,4,byrow = T)
#Data
x1 <- Salary
x2 <- Payroll
if(App==1){
x=x1
meanes=est1[1,]
sdes=est1[2,]
} else if(App==2) {
x=x2
meanes=est2[1,]
sdes=est2[2,]
} else {
print("The value in App is not valid.")
}
#Model
model_code <- '
data {
int n;
real x[n];
vector[4] meanes;
vector[4] sdes;
}
```

```
parameters {
real<lower=0> c;
real<lower=0> k;
real<lower=0> s;
real<lower=0> l;
}
model {
c ~ normal(meanes[1], sdes[1]);
k ~ normal(meanes[2], sdes[2]);
s ~ normal(meanes[3], sdes[3]);
l ~ normal(meanes[4], sdes[4]);
for(i in 1:n){
target += log((l*c*k*s^(-c)*x[i]^(c-1)*(k*log((1+(x[i]/s)^c)))^(-l-1)*
(1+(k*log((1+(x[i]/s)^c)))^(-l))^(-2))*(1+(x[i]/s)^c)^(-1));
}
}'
#Data for Stan
data_list <- list(n = length(x),
x = x)
#Fit Model Stan
fit <- stan(model_code = model_code, data = data_list,
chains = 1, iter = 10000, warmup = 2000,
control = list(adapt_delta = 0.95))
#Result
summary(fit, pars = c("c", "k","s", "l" ))
```

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
