# Peer review of "The Logistic Burr XII Distribution: Properties and Applications to Income Data"

_stats, doi:10.3390/stats6040078_

Round 1

Reviewer 1 Report

Comments and Suggestions for Authors

Reviewer 2 Report

Comments and Suggestions for Authors

I have read the manuscript with great interest: it is a sound statistical endeavor, well structured and presented. It presents a LBXII distribution and its direct application to model income distribution behavior, with possible applications as a competitive model to the classical Weibull distribution, and exponentiated Weibull model. The paper provides a good application following Afify et al 2018, employing two sets of income data and comparing results across different model parameters, MLE and SE, Goodness of fit statistics which highlight how the four parameters LBXII is often a better choice with respect to other distributions, both in terms of MLE and SE.

I do not have major comments and I believe the paper is ready for publication.

Reviewer 3 Report

Comments and Suggestions for Authors

This manuscript introduces the four-parameter logistic Burr XII distribution, a versatile model that can be applied to income distribution and various other domains. The distribution exhibits diverse hazard functions and is characterized by an infinite linear combination of Burr XII densities, with associated mathematical properties and estimation methods explored through simulations and real-world income datasets. Overall, the paper makes a significant contribution to the literature in line with Stats from the theoretical and empirical perspective.

Round 2

Reviewer 1 Report

Comments and Suggestions for Authors

After reading the paper, the author dealt with every big and small comment carefully except for the first comment. The author added Section 4 to the Bayesian method, which is a good effort and adds novelty to the paper, but I want to apply this method to application data (not a simulation study) and put this result on paper even if it is not The best. (We write the theoretical part, where is the application of the method????). If applicable, please add Bayesian code and add a brief  comparison between the two methods.
